# Self-Distilled Reasoner:
# On-Policy Self-Distillation for Large Language Models

**Siyan Zhao**[† 1]  **Zhihui Xie**[2]  **Mengchen Liu**[3]  **Jing Huang**[3]  **Guan Pang**[3]  **Feiyu Chen**[*,‡ 3]  **Aditya Grover**[* 1]

## Abstract

Knowledge distillation improves large language model (LLM) reasoning by compressing the knowledge of a teacher LLM to train smaller LLMs. On-policy distillation advances this approach by having the student sample its own trajectories while a teacher LLM provides dense token-level supervision, addressing the distribution mismatch between training and inference in off-policy distillation methods. However, on-policy distillation typically requires a separate, often larger, teacher LLM and does not explicitly leverage ground-truth solutions available in reasoning datasets. Inspired by the intuition that a sufficiently capable LLM can rationalize external privileged reasoning traces and teach its weaker self, we introduce *On-Policy Self-Distillation* (OPSD), a learning algorithm where a single LLM acts as both teacher and student with different contexts. The teacher policy conditions on privileged information while the student policy sees only the question; training minimizes the per-token divergence between these distributions over the student's own rollouts. We demonstrate the efficacy of our method on multiple mathematical reasoning benchmarks, achieving superior token efficiency compared to reinforcement learning methods and better performance over off-policy distillation methods. Code repo: https://github.com/siyan-zhao/OPSD.

## 1. Introduction

Recent advances in large language models (LLMs) have demonstrated impressive capabilities in reasoning and instruction following. Achieving these capabilities during post-training typically relies on reinforcement learning methods such as Reinforcement Learning with Verifiable Rewards (RLVR) (e.g., GRPO (Shao et al., 2024; Guo et al., 2025; Team et al., 2025; Rastogi et al., 2025; Yu et al., 2025)), supervised fine-tuning (SFT) on high-quality reasoning datasets (Guha et al., 2025; Team et al., 2025; Xiaomi, 2026), or knowledge distillation, where recent work has shown that distillation from advanced teacher models can outperform RL in both performance and training efficiency (Yang et al., 2025; Xiaomi, 2026; Lu & Lab, 2025).

Despite their respective successes, each approach has inherent limitations. RLVR suffers from inefficiencies including: (1) sampling a group of responses per prompt is computationally expensive and can introduce high variance in estimating the true value function; moreover, when all samples are either correct or incorrect, the gradient signal vanishes (Yu et al., 2025; Zhao et al., 2025); and (2) the reward signal is sparse and uniformly applied across all tokens in the generated output, neglecting fine-grained token-level feedback. Supervised fine-tuning suffers from exposure bias and weaker generalization (Agarwal et al., 2024; Chu et al., 2025). Traditional knowledge distillation provides dense token-level supervision from a teacher model but relies on off-policy data (Hinton et al., 2015). Recent advances in on-policy distillation—where a student model samples its own trajectories while a teacher policy provides dense token-level supervision—have demonstrated superior sample efficiency by combining the distributional realism of on-policy training with dense feedback (Agarwal et al., 2024; Lu & Lab, 2025).

While on-policy distillation has shown strong performance, it relies on a distinct teacher model to supervise the student. Given that modern LLMs already exhibit strong reasoning capabilities, we ask this research question: *can a model effectively serve as its own teacher through self-distillation?* Our approach is inspired by human learning: after solving a problem incorrectly, a student can examine the correct solution, rationalize its steps, and identify where their reasoning failed. Prior work has shown that for LLMs, evaluation is often easier than generation (Sun et al., 2024; Naor, 1996). We hypothesize that *rationalization*—explaining a given cor-

---

[*]Equal advising,[†] Work done at UCLA and during Siyan's part-time internship at Meta.,[‡] Work done at Meta. [1]UCLA [2]HKU [3]Meta Superintelligence Labs. Correspondence to: Siyan Zhao <siyanz@g.ucla.edu>.

*Proceedings of the 43rd International Conference on Machine Learning*, Seoul, South Korea. PMLR 306, 2026. Copyright 2026 by the author(s).

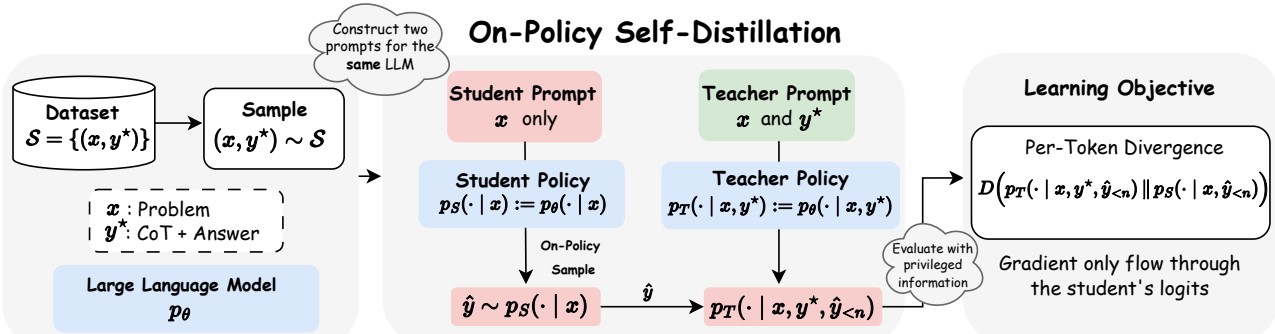

*Figure 1.* **Overview of On-Policy Self-Distillation (OPSD):** Given a reasoning dataset $\mathcal{S} = \{(x_i, y_i^\star)\}_{i=1}^N$, we instantiate two policies from the same LLM: a *student policy* $p_S(\cdot \mid x)$ and a *teacher policy* $p_T(\cdot \mid x, y^\star)$. The student generates an on-policy response $\hat{y} \sim p_S(\cdot \mid x)$. Both policies then evaluate this trajectory to produce next-token distributions $p_S(\cdot \mid x, \hat{y}_{<n})$ and $p_T(\cdot \mid x, y^\star, \hat{y}_{<n})$ at each step $n$. The learning objective minimizes the per-token divergence $D(p_T \| p_S)$ along the student's rollout. The divergence here can be forward KL, reverse KL or JSD. Crucially, gradients backpropagate only through the student's logits, allowing the model to self-distil.

rect answer—is similarly easier than generation. Motivated by this, we instantiate both the teacher and student policies from a single LLM. The teacher policy is provided with privileged information $y^\star$, such as the ground-truth answer or a reference chain-of-thought, while the student policy conditions only on the problem $x$. Concretely, the teacher policy $p_T(\cdot \mid x, y^\star)$ conditions on both the problem and the privileged answer, whereas the student policy $p_S(\cdot \mid x)$ observes only the problem. We preserve the on-policy training paradigm by sampling trajectories $\hat{y}$ exclusively from the student policy, which then receives dense, token-level supervision from the privileged teacher policy.

We therefore propose **On-Policy Self-Distillation (OPSD)**, a framework in which a single model plays both teacher and student roles. The student samples its own trajectories $\hat{y} \sim p_S(\cdot \mid x)$; we then compute the per-token divergence between the student and teacher distributions and minimize it over the student's own rollouts. This formulation (i) uses on-policy supervision (the student's own trajectories), (ii) provides dense per-token feedback, (iii) exploits ground-truth solutions $y^\star$, and (iv) requires no separate teacher model. The learning process is captured by the loss

$$\mathcal{L}_{\text{OPSD}}(\theta) = \mathbb{E}_{(x,y^\star)\sim\mathcal{S}} \, \mathbb{E}_{\hat{y}\sim p_S(\cdot|x)} \sum_{n=1}^{|\hat{y}|}$$
$$D\Big(p_T(\cdot \mid x, y^\star, \hat{y}_{<n}) \, \Big\| \, p_S(\cdot \mid x, \hat{y}_{<n})\Big). \quad (1)$$

In summary, our contributions are as follows:

- We introduce On-Policy Self-Distillation (OPSD), a novel framework that enables a single model to act as both teacher and student, leveraging ground-truth answers to provide dense token-level supervision on student rollouts.
- We introduce a per-token pointwise KL clipping mechanism that stabilizes training and improves performance as

we find stylistic tokens can dominate the training signal of math tokens.
- We evaluate OPSD on three competition-level mathematical reasoning tasks, demonstrating that it matches the performance of GRPO with significantly improved token efficiency and outperform supervised fine-tuning.
- We analyze the impact of different divergence objectives, the effect of student generation length, and student–teacher generation styles.

## 2. Background

### 2.1. Knowledge Distillation for Autoregressive Large Language Models

Knowledge distillation transfers knowledge from a larger teacher model to a smaller student model by training the student to mimic the teacher's behavior (Hinton et al., 2015; Kim & Rush, 2016; Sanh et al., 2019). The core insight is that the teacher's soft probability distribution over classes contains richer information than hard labels alone, as it reveals the teacher's learned similarities between classes. For auto-regressive language models, given a dataset $\mathcal{S} = \{(x, y^\star)\}$ where $x$ denotes an input and $y^\star$ is the corresponding reference output, both teacher $p_T$ and student $p_S$ define token-level distributions over vocabulary $\mathcal{V}$. Traditional supervised distillation minimizes a divergence $D$ between teacher and student distributions averaged over a fixed dataset:

$$\mathcal{L}_{\text{Supervised Distillation}}(\theta) = \mathbb{E}_{(x,y)\sim\mathcal{S}}[D(p_T\|p_S)(y|x)], \quad (2)$$

where $D(p_T\|p_S)(y|x) = \frac{1}{|y|}\sum_{n=1}^{|y|} D(p_T(\cdot|y_{<n},x)\|p_S(\cdot|y_{<n},x))$ measures per-token discrepancy. However, this off-policy approach suffers from distribution mismatch: the student encounters different partial sequences $y_{<n}$ during auto-regressive

| | SFT/Off-Policy Distillation | GRPO | On-Policy Distillation | On-Policy Self-Distillation (Ours) |
|---|---|---|---|---|
| On-Policy Data | ✗ | ✓ | ✓ | ✓ |
| Dense Learning Signal | ✓ | ✗ | ✓ | ✓ |
| Low Sampling Cost | ✓ | ✗ | ✓ | ✓ |
| No External Teacher | ✓ | ✓ | ✗ | ✓ |

*Table 1.* Comparison of training methods for reasoning tasks. On-Policy Self-Distillation (OPSD) combines the advantages of on-policy training with dense feedback without requiring an external teacher model.

generation at inference than those seen during training on the fixed dataset, leading to compounding errors. On-policy distillation (Agarwal et al., 2024; Lu & Lab, 2025; Xu et al., 2024a) addresses this by training the student on its own generated sequences $\hat{y} \sim p_S(\cdot|x)$, obtaining dense token-level feedback from the teacher on these on-policy samples:

$$\mathcal{L}_{\text{On-Policy Distillation}}(\theta) = \mathbb{E}_{x \sim \mathcal{S}}[\mathbb{E}_{\hat{y} \sim p_S(\cdot|x)}[D(p_T \| p_S)(\hat{y}|x)]]. \quad (3)$$

This approach connects distillation to imitation learning (Ross et al., 2011), where the student iteratively improves by learning from the teacher's guidance on its own outputs, combining the on-policy relevance of reinforcement learning with the dense reward signal of supervised learning, thereby mitigating exposure bias while maintaining computational efficiency.

## 2.2. Reinforcement Learning with Verifiable Rewards

Reinforcement learning with verifiable rewards (RLVR) has emerged as a popular approach for post-training large language models, particularly on tasks with easily verifiable outcomes such as mathematics and coding, using algorithms like Proximal Policy Optimization (PPO) (Schulman et al., 2017) and Group Relative Policy Optimization (GRPO) (Shao et al., 2024).

GRPO trains by sampling a group of $G$ responses $\{o_1, o_2, \ldots, o_G\}$ from the current policy $\pi_\theta$ for each problem $x$. Each response $o_i$ receives a binary reward $r_i \in \{0, 1\}$ indicating correctness. The method then assigns advantages to all tokens $k = 1, \ldots, |o_i|$ within response $o_i$ using a group-normalized reward:

$$A_i = \frac{r_i - \text{mean}(\{r_j\}_{j=1}^G)}{\text{std}(\{r_j\}_{j=1}^G)}. \quad (4)$$

This formulation can be understood through the value function lens: $\text{mean}(\{r_j\}_{j=1}^G)$ serves as a $G$-sample Monte Carlo estimate of the value function $V(x)$, while the sparse binary reward $r_i$ represents the (undiscounted) state-action value $Q(x, o_i)$. Critically, all tokens within a response share the same advantage, as the reward signal is provided only

at the sequence level. The GRPO objective incorporates a clipped surrogate loss to moderate policy updates, along with a reverse KL penalty to prevent excessive deviation from a reference policy:

$$\mathcal{L}_{\text{GRPO}}(\theta) = \mathbb{E}_{\substack{x \sim \mathcal{S} \\ o_1, \ldots, o_G \sim \pi_\theta(\cdot|x)}} \left[ \frac{1}{G} \sum_{i=1}^{G} \frac{1}{|o_i|} \sum_{n=1}^{|o_i|} \right.$$
$$\min\left(\rho_i^n A_i, \text{clip}\left(\rho_i^n, 1 - \varepsilon, 1 + \varepsilon\right) A_i\right) \quad (5)$$
$$\left. -\beta D_{\text{KL}}[\pi_\theta(\cdot|x) \| \pi_{\text{ref}}(\cdot|x)] \right]$$

where $\rho_i^n = \frac{\pi_\theta(o_i^n|x, o_i^{<n})}{\pi_{\theta_{\text{old}}}(o_i^n|x, o_i^{<n})}$ is the importance ratio, $\pi_{\theta_{\text{old}}}$ is the policy before the update, and $\varepsilon$ controls the clipping range.

While RLVR methods have demonstrated strong empirical performance, they face two key limitations: (1) the reward signal is sparse, providing only sequence-level feedback rather than token-level guidance on where errors occur, and (2) when all sampled responses receive identical rewards (all correct or all incorrect), the advantages become zero, preventing any policy update despite the computational cost of sampling.

## 3. Methods

### 3.1. Learning from Verifiable Reasoning Dataset

We consider a dataset of problem-solution pairs $\mathcal{S} = \{(x_i, y_i^\star)\}_{i=1}^N$, where each $x_i$ denotes a problem and $y_i^\star$ is the corresponding reference solution, which may include chain-of-thought reasoning. For brevity, we omit the sample index $i$ and use $(x, y^\star)$ to denote a generic sample from the dataset. We can exploit learning signals from this dataset from different ways: Standard supervised fine-tuning (SFT) on $\mathcal{S}$ can be viewed as off-policy distillation/imitation learning using expert trajectories, but it suffers from distribution mismatch between training and inference. Reinforcement learning from verifiable rewards (RLVR), such as GRPO, addresses this by optimizing on-policy samples and assigning binary rewards by comparing generated answers against $y^\star$. However, RLVR is computationally expensive and the

> **Student Prompt**
>
> ```
> Problem:  Find the derivative of f(x) = 3x² + 2x − 5 at x = 2
>
> Answer:
> ```

> **Teacher Prompt**
>
> ```
> Problem:  Find the derivative of f(x) = 3x² + 2x − 5 at x = 2
> Here is a reference solution:
> First find f'(x) = 6x + 2, then evaluate at x = 2:   f'(2) = 6(2) + 2 = 14
> ```
> **After understanding the reference solution, please try to solve this problem using your own approach below:**
> ```
> Answer:
> ```

*Figure 2.* **Prompt example for student and teacher policies.** Both policies share the same parameters $\theta$ but differ in conditioning context. The teacher receives the ground-truth solution $y^\star$ as privileged information before generation. To ensure a natural transition before evaluating the student's rollout, the teacher is prompted to rationalize and generate its own solution. Note that the teacher won't be generating tokens—rationalization is done implicitly through one forward pass.

reward signal is sparse, providing same feedback across all tokens regardless of where errors occur. Alternatively, one can train a process reward model (PRM) to provide dense, token-level feedback during RL. However, acquiring labels for PRM training is prohibitively expensive and difficult to scale (Lightman et al., 2023; Zhang et al., 2025). On-policy distillation works (Agarwal et al., 2024; Xu et al., 2024a; Lu & Lab, 2025) address distribution shift by training on the student's own samples, but require a separate, often larger, teacher model to provide supervision. We instead seek a training signal that is *dense*, *on-policy*, and *does not require external teachers or reward models*. This motivates our On-Policy Self-Distillation approach. We summarize the differences of these methods in Table 1.

### 3.2. On-Policy Self-Distillation

**Motivation: Learning by understanding solutions.** We propose a different perspective inspired by how students learn: when struggling with a problem, rather than extended trial-and-error, a student can examine the solution, understand the reasoning, and internalize the approach. Similarly, if a model has access to the correct answer or reasoning $y^\star$ and is sufficiently capable, it can rationalize the reasoning steps and teach itself—analogous to a student reviewing a solution and retracing why it works. This intuition motivates our framework: we exploit the ground-truth solution $y^\star$ directly as privileged information during training, enabling the model to serve as its own teacher without requiring external reward models or larger teacher models.

**Teacher and student policies.** We instantiate two conditional distributions from the same language model $p_\theta$ by

varying the conditioning context. The *teacher policy* conditions on privileged information—both the problem $x$ and the reference solution $y^\star$:

$$p_T(\cdot \mid x, y^\star) \triangleq p_\theta(\cdot \mid x, y^\star).$$

The *student policy* observes only the problem statement, matching the inference-time condition:

$$p_S(\cdot \mid x) \triangleq p_\theta(\cdot \mid x).$$

Both policies share the same parameters $\theta$ but differ only in their conditioning context. To encourage the teacher to naturally evaluate the student's generation, we add a prompt asking the teacher to generate a new solution after seeing the reference solution as shown in Figure 2. However, the teacher doesn't generate tokens, it only does rationalization implicitly through prefilling.

**On-policy sampling from the student.** Given a problem $x$, the student generates an on-policy response

$$\hat{y} = (\hat{y}_1, \ldots, \hat{y}_{|\hat{y}|}) \sim p_S(\cdot \mid x).$$

Both policies then evaluate this student-generated trajectory. At each position $n$, they induce *next-token* distributions over $y_n \in \mathcal{V}$ conditioned on the same student prefix:

$$p_S(y_n \mid x, \hat{y}_{<n}), \qquad p_T(y_n \mid x, y^\star, \hat{y}_{<n}),$$

where $\hat{y}_{<n} \triangleq (\hat{y}_1, \ldots, \hat{y}_{n-1})$.

**Training objective: Full-vocabulary logit distillation.** We instantiate a *full-vocabulary divergence objective* that

---

**Algorithm 1** On-Policy Self-Distillation (OPSD)

---

**Require:** Reasoning dataset $\mathcal{S} = \{(x_i, y_i^\star)\}_{i=1}^N$; language model $p_\theta$; divergence $D$ (e.g., $\mathrm{JSD}_\beta$)

1: Let $p_S(\cdot \mid x)$ and $p_T(\cdot \mid x, y^\star)$ be the same model $p_\theta$ under different conditioning.
2: **while** not converged **do**
3:      Sample a minibatch $\mathcal{B} \subset \mathcal{S}$
4:      **for all** $(x, y^\star) \in \mathcal{B}$ **do**
5:          Sample on-policy response $\hat{y} \sim p_S(\cdot \mid x)$
6:          Compute the token-wise divergence along the student rollout:

$$\ell(x, y^\star) \leftarrow D(p_T \| p_S)(\hat{y} \mid x) = \frac{1}{|\hat{y}|} \sum_{n=1}^{|\hat{y}|} D\big(p_T(\cdot \mid \hat{y}_{<n}, x, y^\star) \,\big\|\, p_S(\cdot \mid \hat{y}_{<n}, x)\big)$$

7:      Calculate loss $\mathcal{L}_{\mathrm{OPSD}}(\theta) \leftarrow \frac{1}{|\mathcal{B}|} \sum_{(x,y^\star) \in \mathcal{B}} \ell(x, y^\star)$ and update $\theta$

---

matches the teacher and student next-token distributions at each position. Given a student-generated sequence $\hat{y}$, define the trajectory-averaged, token-wise divergence

$$D(p_T \| p_S)(\hat{y} \mid x) \triangleq \frac{1}{|\hat{y}|} \sum_{n=1}^{|\hat{y}|} D\bigg(p_T(\cdot \mid x, y^\star, \hat{y}_{<n}) \tag{6}$$
$$\big\| \, p_S(\cdot \mid x, \hat{y}_{<n})\bigg),$$

where $p_S(\cdot \mid x, \hat{y}_{<n})$ and $p_T(\cdot \mid x, y^\star, \hat{y}_{<n})$ denote distributions over the next token $y_n \in \mathcal{V}$. Here, $D$ can be any distribution divergence measure such as the *generalized Jensen-Shannon divergence* $\mathrm{JSD}_\beta$, defined for a weight $\beta \in [0, 1]$ as:

$$\mathrm{JSD}_\beta(p_T \| p_S) = \beta D_{KL}(p_T \| m) + (1 - \beta) D_{KL}(p_S \| m) \tag{7}$$

where $m = \beta p_T + (1 - \beta) p_S$ is the interpolated mixture distribution. This full-vocabulary formulation provides dense, token-level feedback: the teacher, informed by $y^\star$, exposes the student to the entire distribution over plausible next tokens and guides it toward reasoning paths that lead to the correct answer.

We minimize the expected divergence between teacher and student over on-policy student samples:

$$\mathcal{L}(\theta) = \mathbb{E}_{(x,y^\star) \sim \mathcal{S}} \big[\mathbb{E}_{\hat{y} \sim p_S(\cdot \mid x)} \big[D(p_T \| p_S)(\hat{y} \mid x)\big]\big]. \tag{8}$$

Gradients are backpropagated only through the student policy $p_S$, while the teacher $p_T$ acts as a fixed full-distribution target conditioned on privileged information $(x, y^\star)$.

**Per-Token Pointwise Divergence Clipping.** In our experiments, we observe that token-level divergence is highly skewed across vocabulary entries: a small subset of stylistic tokens exhibits much higher divergence than mathematically meaningful tokens (see Table 7). This imbalance causes the

training signal to be dominated by stylistic patterns. To address this, we apply pointwise clipping to the vocabulary-level divergence contributions. Let $D_f(p_T \| p_S)$ denote an $f$-divergence. At each token position $n$ and vocabulary entry $v$, define:

$$\ell_{n,v}^{(f)} = p_T(v \mid \cdot) \, f\left(\frac{p_S(v \mid \cdot)}{p_T(v \mid \cdot)}\right).$$

We compute the clipped divergence:

$$D_{\mathrm{clip}}^{(f)}(p_T \| p_S) = \frac{1}{|\hat{y}|} \sum_{n=1}^{|\hat{y}|} \sum_{v \in \mathcal{V}} \min(\ell_{n,v}^{(f)}, \tau).$$

**Alternative objective: Sampled-token distillation through policy gradient.** Following recent on-policy distillation methods (Lu & Lab, 2025), we form a sampled-token reward signal (a reverse-KL signal on sampled actions) and optimize with policy gradient. For each position $n$ in a sampled sequence $\hat{y}$, define the advantage term

$$A_n(x, \hat{y}) = \log p_T(\hat{y}_n \mid x, y^\star, \hat{y}_{<n}) - \log p_S(\hat{y}_n \mid x, \hat{y}_{<n}),$$

and optimize the policy-gradient-style objective

$$\mathcal{L}(\theta) = -\mathbb{E}_{(x,y^\star) \sim \mathcal{S}} \bigg[\mathbb{E}_{\hat{y} \sim p_S(\cdot \mid x)} \bigg[\frac{1}{|\hat{y}|} \sum_{n=1}^{|\hat{y}|} A_n(x, \hat{y}) \tag{9}$$
$$\times \log p_S(\hat{y}_n \mid x, \hat{y}_{<n})\bigg]\bigg].$$

$A_n(x, \hat{y})$ is treated as a constant with respect to $\theta$ (i.e., gradients do not flow through the advantage), so that gradients take the usual policy-gradient form $A_n \nabla_\theta \log p_S$. Compared to the full-vocabulary divergence objective, this on-policy shaping objective operates only on sampled tokens, using the teacher's log-probabilities to provide dense, trajectory-level shaping signals without explicitly matching the full distribution at each step.

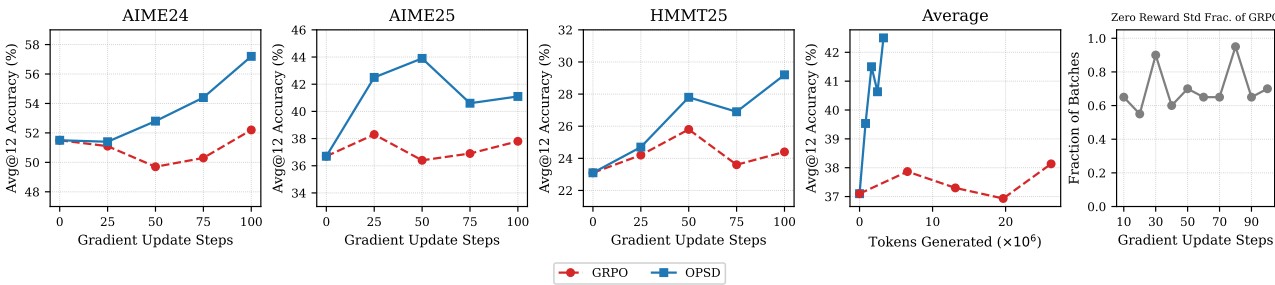

*Figure 3.* **Token Efficiency of OPSD.** We compare OPSD and GRPO on Qwen3-1.7B under the same effective training batch size, reporting Avg@12 accuracy with training steps and total tokens generated. Generation is capped at 1024 tokens for OPSD and 16k for GRPO. At the same number of training steps, OPSD uses significantly fewer tokens but outperforms GRPO on all benchmarks. Despite sampling more tokens, GRPO only receives a binary outcome reward, and stagnates due to reward diversity collapse (rightmost plot): more than half of its batches have zero reward standard deviation within 100 steps, yielding no gradient signal. OPSD sidesteps this disadvantage of outcome-based rewards by learning from a dense distillation loss even with fewer generated tokens.

**OPSD as dense-reward policy gradient and comparison to STaR.** The objective in Equation (9) can be seen as policy gradient with dense, token-level rewards. In Appendix Section D, we formalize this and contrast with STaR (Zelikman et al., 2022), a closely related method that also uses the same model to generate reasoning traces, then performs rejection sampling followed by SFT on correct traces. This procedure can be viewed as policy gradient with a sequence-level binary reward that assigns identical credit to all tokens and vanishes when samples are incorrect. In contrast, OPSD provides feedback at every token position regardless of final-answer correctness.

## 4. Experiments

We conduct comprehensive experiments to answer the following research questions:

(1) How does OPSD compare to SFT and GRPO in reasoning performance and sample efficiency? (§4.2)
(2) How does per-token pointwise KL clipping in OPSD help stabilizing training? (§4.3.3)
(3) What is the effect of generation style, generation length on performance? (§4.3.4)
(4) Does full-vocabulary logit distillation provide benefits over sampled-token policy gradient? (§4.3.5)

### 4.1. Experimental Setup

**Models and datasets.** We experiment with the Qwen3 (Team, 2025b) model family at three scales: Qwen3-1.7B, Qwen3-4B, and Qwen3-8B, using the instruct-tuned versions. For training data, we use the mathematical reasoning subset of OpenThoughts (Guha et al., 2025), sampling up to 30K problem-solution pairs with chain-of-thought reasoning. We evaluate on competition-level mathematics benchmarks including AIME 2024, AIME 2025, HMMT 2025.

**Baselines.** We compare against two methods trained on the same dataset: (1) **SFT**, standard supervised fine-tuning on expert trajectories, which can be seen as off-policy distillation from a more powerful LLM that generated the reasoning traces; (2) **GRPO** (Shao et al., 2024), group relative policy optimization with binary outcome rewards verified against ground-truth answers. The max generation length is set to 16k.

**Implementation details.** We fix the teacher policy to be the initial policy, rather than the currently updating learning policy, as we find this helps stabilize training and implicitly acts as regularization to prevent excessive deviation from the initial policy. We use full-vocabulary logit distillation in our experiments. All experiments are conducted on A100 or H100 GPUs with LoRA (Hu et al., 2022). More experimental details are in Appendix B.

### 4.2. Main Results

Table 2 reports results on competition-level mathematical reasoning benchmarks. OPSD consistently outperforms SFT and improves over the base model across all scales, matching or exceeding GRPO in every setting. Notably, OPSD achieves these gains using only a single rollout per problem and converges within 100 steps, with each problem requiring only 1024 sampled tokens, whereas GRPO requires 8 rollouts of 16k tokens each and may exhibit performance degradation in later steps due to entropy collapse—with most of reward standard deviations within a group being zero under this OpenThoughts dataset, yielding no learning signal and wasting sampling budget. We also observe consistent performance degradation under SFT across tasks and model scales when trained on the same dataset, which we attribute to the concise reasoning style of the ground truth solutions which has reduced reasoning lengths at test time. We attribute OPSD's token efficiency to dense token-level supervision from the teacher distribution, and we hypoth-

*Table 2.* Performance comparison on mathematical reasoning benchmarks for Qwen3 models. We report Avg@12 under the sampling configuration recommended in the Qwen3 blog (temperature 1.0, maximum generation length 38k); full details are provided in Table 10. For OPSD, we evaluate checkpoints every 20 steps up to 100 steps and report the best score. For GRPO, we report the peak performance within 500 training steps, though we find GRPO performance to decrease for some tasks due to entropy collapse in later steps. For SFT, we train on the same number of samples as OPSD. SFT performance degrades due to fine-tuning on concise reasoning solutions and reduces generation length at test time, whereas OPSD transforms them into dense learning signal through rationalization.

| Method | AIME24 | AIME25 | HMMT25 | Average |
|---|---|---|---|---|
| *Qwen3-8B* | | | | |
| Base (Instruct) | 75.8 | 65.6 | 43.9 | 61.8 |
| + SFT | 72.3 | 64.2 | 42.9 | 59.8 |
| + GRPO | 76.4 | 68.9 | **46.7** | 64.0 |
| + OPSD | **77.8** | **70.8** | 45.8 | **64.8** |
| *Qwen3-4B* | | | | |
| Base (Instruct) | 74.9 | 66.4 | 42.2 | 61.2 |
| + SFT | 70.2 | 62.3 | 43.4 | 58.6 |
| + GRPO | 75.6 | 68.1 | 44.4 | 62.7 |
| + OPSD | **76.4** | **68.3** | **46.1** | **63.6** |
| *Qwen3-1.7B* | | | | |
| Base (Instruct) | 51.5 | 36.7 | 23.1 | 37.1 |
| + SFT | 48.4 | 36.3 | 22.7 | 35.8 |
| + GRPO | 51.1 | 38.3 | 23.7 | 37.7 |
| + OPSD | **57.2** | **43.9** | **29.2** | **43.4** |

esize that earlier tokens may contribute more to effective distillation as they could represent more critical branching points in the reasoning process.

As shown in Figure 3, OPSD achieves higher token learning efficiency within 100 steps of training as compared to GRPO. Within 100 steps, GRPO's performance stagnates with less learning signal when the outcome reward within as sampling group remains the same, leading to zero gradient. These results suggest that OPSD may extract learning signal from the same reasoning datasets more efficiently than both GRPO and SFT, while substantially reducing training time.

### 4.3. Ablation Studies & Discussions

In this section, we conduct extensive ablations to study key design choices in OPSD, including (1) the divergence objective, (2) the generation styles of the student and teacher (e.g., thinking-mode on/off), (3) the effect of per-token KL clipping, (4) the impact of student generation length, and (5) comparison between full-vocabulary logit distillation with sampled-token distillation.

#### 4.3.1. EFFECT OF DIVERGENCE OBJECTIVE

A key design choice in OPSD is the divergence used for per-token distribution matching between the privileged teacher and the student. We compare forward KL, reverse KL, and

JSD on AIME25 with Qwen3-1.7B in Table 3. All objectives are evaluated under the same pointwise clipping scheme for stability. Forward KL consistently yields the strongest gains, improving performance from 36.7 to 43.9 at step 50 and remaining above the baseline at step 100. In contrast, reverse KL and JSD provide limited or negative improvements. We therefore adopt forward KL in all remaining experiments.

*Table 3.* Comparison of divergence objectives on AIME25 with Qwen3-1.7B. We report Avg@12 at different training steps. Forward KL significantly improves performance over the base model, while reverse KL and JSD ($\beta = 0.5$) show limited or negative gains.

| Method | Base | Step 50 | Step 100 |
|---|---|---|---|
| Forward KL ($\mathrm{KL}(p_T \parallel p_S)$) | 36.7 | **43.9** | **41.1** |
| Reverse KL ($\mathrm{KL}(p_S \parallel p_T)$) | 36.7 | 37.5 | 35.0 |
| JSD ($\beta = 0.5$) | 36.7 | 36.9 | 39.0 |

#### 4.3.2. EFFECT OF GENERATION STYLES AND PER-TOKEN KL CLIPPING

Another key design choice in OPSD is the generation style of the student and teacher models, as it determines both which tokens the student learns from and the style of supervision provided by the teacher. Qwen3 models support two generation modes: *Thinking Mode on* (TM-on), in which

the model produces self-reflective chain-of-thought tokens, and *Thinking Mode off* (TM-off), in which it generates responses directly. To determine which combination yields the most effective learning signal, we analyze the forward KL divergence $\text{KL}(p_T \| p_S)$ across all four student/teacher mode pairings, categorizing tokens into three groups: *math* (numerals, operators, and mathematical keywords), *style* (reasoning connectives), and *other*. Table 7 reports the mean per-token KL within each category.

Across all model sizes, the TM-off student paired with a TM-on teacher yields the largest KL on math tokens, indicating stronger supervision on mathematically relevant tokens. The reported KL values correspond to the expected divergence over the vocabulary at each position; as shown in Table 7, this expectation is highly skewed, with stylistic tokens contributing disproportionately large values. This motivates our use of pointwise clipping to control such heavy-tailed contributions. Empirically, this configuration achieves the best downstream performance. We therefore adopt the TM-off student / TM-on teacher configuration.

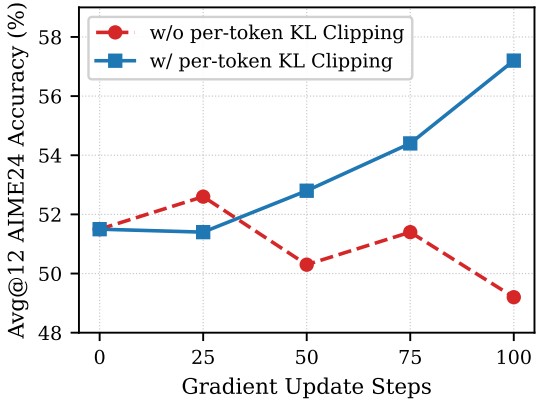

*Figure 4.* Effect of Per-Token pointwise KL Clipping on Qwen3-1.7B evaluated on AIME24. Clipping prevents performance collapse.

### 4.3.3. EFFECT OF PER-TOKEN POINTWISE CLIPPING

As shown in Table 7, stylistic tokens can exhibit higher KL divergence than math-related tokens, causing them to dominate the training signal. We mitigate this issue using per-token pointwise clipping. As shown in Figure 4 for Qwen3-1.7B, clipping stabilizes training and prevents performance degradation, which is particularly important given that OPSD converges rapidly within a hundred steps of training. Note that the training loss can be negative due to pointwise clipping before aggregating.

### 4.3.4. EFFECT OF GENERATION LENGTH

Since our objective operates at the token level (Eq. 6), the number of generated tokens per sample directly determines

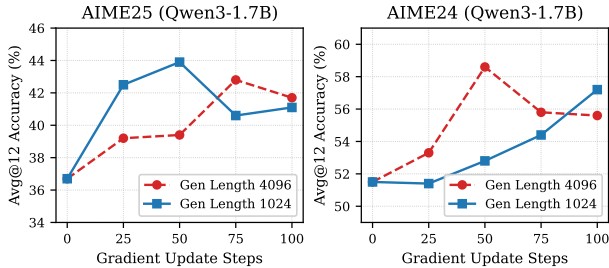

*Figure 5.* Effect of Generation Length on Qwen3-1.7B. We compare student generation length of 1024 vs 4096 on AIME25 and AIME24.

the amount of supervision signal available to the student. Longer sequences expose the student to more teacher feedback, but they also increase computational cost and may introduce noisy or uninformative continuations. To study this trade-off, we conduct an ablation on Qwen3-1.7B by varying the generation length of on-policy sampled student responses among 1024 and 4096 tokens and use full-vocabulary logit distillation. As shown in Figure 5, increasing the generation length does not lead to consistent improvements across either task. We attribute this to early tokens being more critical for learning: as the student generation grows longer, later tokens become increasingly predictable to the teacher when conditioned on a sufficiently long student prefix so less penalties are applied to later tokens. This phenomenon is also noted in (Lu & Lab, 2025).

### 4.3.5. LEARNING OBJECTIVE COMPARISON: FULL VOCABULARY LOGITS DISTILLATION VS. SAMPLED-TOKEN DISTILLATION

Our objective in Eq. 6 is defined as a per-token discrepancy between the teacher and student *distributions*. In practice, OPSD can instantiate this objective in two ways. (1) **Full-vocabulary logit distillation** (as in GKD (Agarwal et al., 2024)): for each token position, we compute $D(p_T \| p_S)$ over the entire vocabulary via a full softmax, yielding a proper token-level $f$-divergence between the two policies. (2) **Sampled-token advantage policy-gradient objective** (as in the on-policy distillation method of Lu & Lab (2025)): we evaluate teacher and student log-probabilities only at the token actually sampled by the student, $\hat{y}_n$, and use the reverse-KL term as a scalar advantage inside a policy-gradient-style loss. Thus, the first variant directly matches full token distributions, whereas the second optimizes an on-policy RL objective shaped by the teacher's log-probabilities rather than a full-distribution divergence. We compare these variants on Qwen3-4B using a 2048-token generation budget during distillation. Table 4 summarizes the results. The full-vocabulary divergence objective provides a consistent gain over the sampled-token objective. This suggests that exposing the student to the full teacher distribution offers

*Table 4.* Ablation on divergence computation strategies for OPSD on Qwen3-4B with 2048 generation length for distillation. We report pass@8 accuracy on AIME25 and HMMT25. Full-distribution objectives (logit distillation) outperform sampled-token objectives.

| Method Variant | AIME25 | HMMT25 |
|---|---|---|
| OPSD w/ Full-vocabulary logit distillation (Agarwal et al., 2024) | **84.1** | **60.0** |
| OPSD w/ Sampled-token distillation (Lu & Lab, 2025) | 82.1 | 57.3 |

richer supervision than relying solely on per-token on-policy shaping. However, the full-vocabulary computation incurs higher peak memory usage due to storing vocabulary-sized logits at every position, indicating a trade-off between performance and efficiency.

## 5. Related Work

**LLM Self-Training.** Our work connects to a line of research showing that LLMs can improve by generating and exploiting their own supervision signals (Allen-Zhu & Li, 2020; Xu et al., 2024b; Chen et al., 2024; Wang et al., 2023; Sun et al., 2023; Yuan et al., 2024; Yang et al., 2024). Closest in spirit is *context distillation* (Snell et al., 2022), which uses the same underlying model as both teacher and student by providing the teacher with privileged context and then SFT the student on the teacher's *generated* outputs without context. This can be viewed as *off-policy*, where the learning signal is a discrete token sequence. In the reasoning domain, ReST (Gulcehre et al., 2023) and STaR (Zelikman et al., 2022) similarly rely on iterative self-training loops—generate rationales conditioned on hints or answers, filter by rewards or ground-truth answers, and fine-tune on successful trajectories—again yielding hard distillation; Mitra & Ulukus (2025) extends this to soft distillation. In-context editing (Qi et al., 2025) does on-policy sample from student and shows that *context-induced* knowledge can be internalized via soft distillation by minimizing divergences and demonstrates this in knowledge editing settings. OPSD differs from these approaches in that we perform *on-policy, soft distillation* on the student's own rollouts for reasoning tasks: the teacher's supervision is per-token distribution matching rather than generating a rationale for SFT. OPSD frames reasoning improvement as learning a conditional distribution induced jointly by the dataset's ground-truth solutions and the model's own reasoning ability. Concurrently, SDPO (Hübotter et al., 2026) explored similar algorithm with environment feedbacks as priviledged information and SDFT (Shenfeld et al., 2026) explored on-policy self-distillation on continual learning tasks.

**On-Policy Distillation** methods train a student model directly on trajectories sampled from its own policy, while a teacher model provides per-token guidance through KL-based regularization or related objectives (Agarwal et al., 2024; Xu et al., 2024a; Gu et al., 2024; Lu & Lab, 2025;

Xiaomi, 2026; Yang et al., 2025). These approaches mitigate distribution shift by optimizing directly on the student's visitation distribution, but they typically rely on a distinct and often larger teacher model. In this work, we explore whether an LLM can teach itself by conditioning on more privileged answer information and leveraging its own reasoning capability to guide a weaker version of itself toward improved reasoning. On-policy training paradigms are also widely used in robotics and deep reinforcement learning, such as DAgger (Ross et al., 2011), where a human teacher provides corrective supervision on the states visited by the student policy.

**Improving LLM Reasoning through SFT and RL.** SFT and RL are two primary methods for improving LLM reasoning ability. SFT on high-quality reasoning traces has demonstrated strong performance (Yu et al., 2023; LI et al., 2024; Paster et al., 2023; Team, 2025a; Ye et al., 2025; Muennighoff et al., 2025; Zhou et al., 2023). However, prior work shows that SFT can rely on memorization rather than robust generalization (Chu et al., 2025). In contrast, RL optimizes directly for outcome-based objectives can exhibit better generalization (Huan et al., 2025). More recent algorithms such as GRPO (Guo et al., 2025; Shao et al., 2024) enable scalable RL by estimating advantages from group-level rewards without requiring an explicit critic as in PPO (Schulman et al., 2017). Building on this line of work, a growing body of research highlights the effectiveness of RLVR for reasoning tasks (Yu et al., 2025; Liu et al., 2025; Yue et al., 2025; An et al., 2025; Zheng et al., 2025).

## 6. Conclusion

We introduced On-Policy Self-Distillation (OPSD), a simple yet effective framework for post-training large language models on reasoning tasks. The intuition behind OPSD is that a sufficiently capable reasoning LLM can teach itself when it has access to privileged information about the answer to a reasoning problem, utilizing its own rationalization ability to grade its weaker self without access to the ground truth. We experimentally demonstrated that OPSD achieves better performance than off-policy distillation/SFT, and performs on par with or better than GRPO, while exhibiting significantly better sample efficiency than GRPO.

# Impact Statement

This paper presents work whose goal is to advance the field of machine learning. Our method improves the efficiency of training language models for reasoning tasks, reducing computational costs compared to existing reinforcement learning approaches. We do not foresee specific negative societal consequences.

# Acknowledgements

AG was supported by NSF CAREER Grant # 2341040 and a Schmidt AI 2050 Fellowship.

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

# A. Limitations and Future Directions

Due to computational constraints, our experiments are limited to models up to 8B parameters. It remains an open question whether this trend continues at scales beyond 8B parameters and if larger LLMs brings more stability with OPSD as we have observed instability and performance drop in later training steps. Several promising directions warrant further investigation. First, our current framework does not explicitly leverage correctness verification of generated answers; incorporating such signals could provide additional learning objectives and more stability beyond distribution matching. Finally, problem difficulty plays a crucial role in self-distillation: if reasoning problems exceed the model's comprehension threshold, the teacher policy cannot provide meaningful supervision even with access to ground-truth solutions. This suggests that curriculum learning strategies—gradually increasing problem difficulty as the model improves—could enhance training effectiveness. Exploring adaptive curricula that maintain problems at the frontier of model capabilities represents an important direction for scaling OPSD to more challenging reasoning tasks.

# B. Experimental Details

**Non-thinking mode results.**    We further evaluate a fully non-thinking configuration on the Qwen3 family, where both the student and teacher operate with Qwen3 thinking mode disabled during training, and evaluation is also conducted under non-thinking inference. As shown in Table 5, OPSD improves over the corresponding non-thinking base model in most settings, with the largest gains observed on Qwen3-8B: +23.3 on AIME24, +15.3 on AIME25, and +7.5 on HMMT25. The gains are not uniform across scales: the 8B model benefits more substantially than the 4B and 1.7B models.

*Table 5.* Fully non-thinking OPSD results across model scales. Both student and teacher use Qwen3 non-thinking mode during training, and evaluation is also performed with non-thinking inference. We report Avg@12 accuracy. For each model and benchmark, full results across steps are shown in Table 6.

| Model | AIME24 | | | AIME25 | | | HMMT25 | | | Avg. Performance Gain |
|---|---|---|---|---|---|---|---|---|---|---|
| | Base | OPSD | $\Delta$ | Base | OPSD | $\Delta$ | Base | OPSD | $\Delta$ | |
| Qwen3-1.7B | 11.9 | **15.0** | +3.1 | 9.2 | 6.2 | -3.0 | 5.0 | **5.8** | +0.8 | **+0.3** |
| Qwen3-4B | 23.1 | **31.1** | +8.0 | 21.4 | 21.1 | -0.3 | 10.8 | **16.4** | +5.6 | **+4.4** |
| Qwen3-8B | 26.4 | **49.7** | +23.3 | 19.7 | **35.0** | +15.3 | 10.8 | **18.3** | +7.5 | **+15.4** |

*Table 6.* Full step-wise results for fully non-thinking OPSD on Qwen3 models. Both student and teacher use Qwen3 non-thinking mode during training, and evaluation is also performed with non-thinking inference. We report Avg@12 accuracy. The best checkpoint for each model and benchmark is used in Table 5.

| Model | Benchmark | Base | Step 50 | Step 75 | Step 100 |
|---|---|---|---|---|---|
| | AIME24 | 11.9 | **15.0** | 13.9 | 12.5 |
| Qwen3-1.7B | AIME25 | **9.2** | 6.2 | 8.3 | 8.1 |
| | HMMT25 | 5.0 | **5.8** | 5.0 | 5.4 |
| | AIME24 | 23.1 | 20.3 | 27.5 | **31.1** |
| Qwen3-4B | AIME25 | **21.4** | **21.4** | 20.8 | 21.1 |
| | HMMT25 | 10.8 | 11.1 | 13.1 | **16.4** |
| | AIME24 | 26.4 | **49.7** | 45.3 | 38.3 |
| Qwen3-8B | AIME25 | 19.7 | **35.0** | 26.9 | 27.5 |
| | HMMT25 | 10.8 | **18.3** | 17.5 | 15.3 |

We provide the training and evaluation configurations for our SFT, GRPO and OPSD experiments in Tables 9, 8 and 10. Note that we adopt the Thinking-Mode-off student / Thinking-Mode-on teacher configuration for main OPSD experiments. For more experiment details, please refer to our released training code in https://github.com/siyan-zhao/OPSD.We didn't conduct tuning for the clipping parameter $\tau$, optimizing this hyperparameter may yield further performance gains within the same 100-step budget for larger models.

All experiments were conducted using 8 A100 or H100 GPUs with gradient checkpointing and Flash Attention 2 for memory efficiency. We use the AdamW (Loshchilov & Hutter, 2017) optimizer and bfloat16 precision for all training runs. For OPSD, unless otherwise stated, we used full-vocabulary logit distillation.

*Table 7.* **Per-token KL divergence by token category across generation styles.** Mean per-token KL divergence broken down by token category (see Appendix C for detailed definitions), averaged over 10 problems. Thinking Mode OFF/ON indicates whether the student or teacher LLM's prompt format enables thinking mode. We find when student's generation's thinking mode is off and when the teacher's thinking mode is on, the KL signal on math related tokens are the highest. And we choose this setup for our experiments.

| Student | Teacher | Qwen3-1.7B | | | Qwen3-4B | | | Qwen3-8B | | |
|---|---|---|---|---|---|---|---|---|---|---|
| | | Style | Math | Other | Style | Math | Other | Style | Math | Other |
| TM-off | TM-off | 0.68 | 0.12 | 0.11 | 0.61 | 0.06 | 0.10 | 0.56 | 0.05 | 0.11 |
| TM-on | TM-off | 0.51 | 0.10 | 0.17 | 0.41 | 0.05 | 0.18 | 0.33 | 0.05 | 0.15 |
| TM-on | TM-on | 0.51 | 0.09 | 0.08 | 0.50 | 0.04 | 0.09 | 0.42 | 0.04 | 0.08 |
| **TM-off** | **TM-on** | **0.85** | **0.14** | **0.25** | **0.92** | **0.10** | **0.29** | **0.79** | **0.06** | **0.25** |

*Table 8.* Training Configuration for GRPO and OPSD

| Parameter | GRPO | OPSD |
|---|---|---|
| Learning Rate | $5 \times 10^{-6}$ | $5 \times 10^{-6}$ |
| Effective Batch Size | 32 | 32 |
| LoRA Rank ($r$) | 64 | 64 |
| LoRA Alpha ($\alpha$) | 128 | 128 |
| LoRA Target Modules | q_proj, k_proj, v_proj, o_proj, gate_proj, up_proj, down_proj | |
| Max Completion Length | 16,000 | 1024 |
| Number of Generations per Prompt | 8 | 1 |
| Sampling Temperature | 1.2 | 1.1 |
| KL Coefficient ($\beta$) | 0.0 | – |
| Training Steps | 500 | 100 |

## C. Token Category Definitions

We categorize tokens into *style* and *math* groups using predefined keyword lists. These keyword sets are used to analyze the per-token KL divergence stylistic tokens and mathematical knowledge tokens as in Section 4.3.1.

**Style Tokens.** maybe, perhaps, probably, possibly, let, okay, ok, alright, hmm, wait, because, since, so, thus, hence, therefore, but, however, although, though, yet, or, alternatively, instead, otherwise, actually, really, just, simply, basically, very, quite, pretty, rather, fairly, now, then, next, first, second, finally, try, see, check, note, recall, think, idea, strategy, approach, method, way, would, could, should, might, can, huge, large, big, small, tiny, interesting, tricky, complex, simple.

**Math Tokens.** exponential, exponent, power, powers, base, logarithm, logarithms, log, ln, compare, comparing, comparison, less, equal, larger, smaller, greater, factor, factors, prime, divisible, equation, expression, formula, inequality, rational, irrational, real, integer, coefficient, variable, constant, sum, product, difference, quotient, fraction, denominator, numerator, root, square, cube, nth, maximum, minimum, optimize, bound.

## D. Policy-Gradient Interpretation of OPSD and Comparison to STaR

Our OPSD objective in Equation (9) can be interpreted as a policy-gradient update with a *dense, token-level* reward signal derived from privileged information. In this section, we show: (1) OPSD can be seen as a dense-reward policy gradient, and (2) we contrast OPSD with STaR, demonstrating that STaR's learning signal is *sequence-level* while OPSD is *token-level*.

*Table 9.* Training Configuration for SFT.

| Parameter | SFT |
|---|---|
| Learning Rate | $5 \times 10^{-6}$ |
| Effective Batch Size | 32 |
| LoRA Rank ($r$) | 64 |
| LoRA Alpha ($\alpha$) | 128 |
| LoRA Target Modules | q_proj, k_proj, v_proj, o_proj, gate_proj, up_proj, down_proj |
| Max Sequence Length | 16000 |
| Number of Training Step | 100 |

*Table 10.* Evaluation Parameters.

| Parameter | Value |
|---|---|
| Max New Tokens | 38912 |
| Thinking Mode | Enabled |
| Top-p | 0.95 |
| Top-k | -1 |
| Min-p | 0.0 |
| Presence Penalty | 0.0 |
| Samples per Prompt | 12 |
| Temperature | 1.0 |

### D.1. STaR as Sequence-Level Policy-Gradient

STaR (Zelikman et al., 2022) can be viewed as an approximation to an RL-style policy gradient objective. The language model $p_\theta$ induces a joint distribution over rationale $r$ and answer $y$:

$$p_\theta(r, y \mid x) = p_\theta(r \mid x)\, p_\theta(y \mid x, r),$$

where the model first samples a latent rationale $r$ before predicting the final answer $y$. Given an indicator reward $R(y) = \mathbf{1}(y = y^\star)$, the expected return across the dataset $\mathcal{S} = \{(x_i, y_i^\star)\}_{i=1}^N$ is

$$J_{\text{STaR}}(\theta) = \sum_{i=1}^{N} \mathbb{E}_{(r,y) \sim p_\theta(\cdot \mid x_i)} \big[\mathbf{1}(y = y_i^\star)\big]. \tag{10}$$

Applying the log-derivative trick yields a policy gradient:

$$\nabla_\theta J_{\text{STaR}}(\theta) = \sum_{i=1}^{N} \mathbb{E}_{(r,y) \sim p_\theta(\cdot \mid x_i)} \Big[\mathbf{1}(y = y_i^\star)\, \nabla_\theta \log p_\theta(r, y \mid x_i)\Big]. \tag{11}$$

Note that the indicator function discards the gradient for all sampled rationales that do not lead to the correct answer $y_i^\star$: this corresponds to the filtering step in STaR.

One limitation is that STaR's reward is *sequence-level*: the binary indicator $\mathbf{1}(y = y^\star)$ provides the same signal to all tokens in a trajectory, offering no intermediate credit assignment. When all sampled trajectories are all incorrect, the learning signal vanishes.

### D.2. OPSD as Dense-Reward Policy Gradient

The sampled-token objective in Equation (9) can also be viewed as a policy-gradient method, but with a token-level reward. Fix a training pair $(x, y^\star)$ and let the student generate a trajectory $\hat{y} \sim p_S(\cdot \mid x)$. At each position $n$, define the per-token reward:

$$r_n(x, \hat{y}) \triangleq \log p_T(\hat{y}_n \mid x, y^\star, \hat{y}_{<n}) - \log p_S(\hat{y}_n \mid x, \hat{y}_{<n}).$$

This reward measures how much the privileged teacher prefers the sampled token $\hat{y}_n$ relative to the student. As stated in the main text, we treat $r_n$ (equivalently, the advantage $A_n$) as a constant with respect to $\theta$ when computing gradients—that is, we stop gradients through both $p_T$ and $p_S$ in the reward computation. Under this treatment, the gradient of Equation (9) takes the standard policy-gradient form:

$$\nabla_\theta \mathcal{L}(\theta) = -\mathbb{E}_{(x,y^\star)\sim\mathcal{S}} \left[ \mathbb{E}_{\hat{y}\sim p_S(\cdot|x)} \left[ \frac{1}{|\hat{y}|} \sum_{n=1}^{|\hat{y}|} r_n(x,\hat{y}) \, \nabla_\theta \log p_S(\hat{y}_n \mid x, \hat{y}_{<n}) \right] \right],$$

which corresponds to maximizing the expected per-token reward along on-policy student rollouts:

$$J_{OPSD}(\theta) = \mathbb{E}_{(x,y^\star)\sim\mathcal{S}} \left[ \mathbb{E}_{\hat{y}\sim p_S(\cdot|x)} \left[ \frac{1}{|\hat{y}|} \sum_{n=1}^{|\hat{y}|} r_n(x,\hat{y}) \right] \right].$$

This reward is dense: it provides a learning signal at every token position, regardless of whether the final answer is correct.

**Comparison.** Both STaR and OPSD can be understood as policy-gradient methods, but their reward structures differ fundamentally. STaR uses a sequence-level indicator $\mathbf{1}(y = y^\star)$ that assigns the same signal to all tokens; when all sampled trajectories are incorrect, the learning signal vanishes entirely. In contrast, OPSD provides a token-level reward $r_n$ at every position, enabling fine-grained credit assignment even when the final answer is wrong.

