# OpenReview forum: "Self-Distilled Reasoner: On-Policy Self-Distillation for Large Language Models"
_ICML.cc/2026/Conference — ICML 2026 regular_

### Official Review · Reviewer_U4bM · 2026-03-12

**Soundness:** 2
**Presentation:** 4
**Significance:** 3
**Originality:** 2
**Overall Recommendation:** 3
**Confidence:** 3

**Summary:**

The paper introduces a new on-policy distillation framework, On-Policy Self-Distillation (OPSD), in which the teacher and student are identical models with differently conditioned prompts (with and without answer). The method doesn’t require a stronger teacher model and does not explicitly rely on grounding-truth reasoning. The paper reports that the model improves the reasoning performance over other baselines (e.g., SFT and GRPO-trained models), on various math reasoning dataset. Importantly, they claim that the model achieves comparable performance with GRPO with much fewer generated tokens.

**Compliance With Llm Reviewing Policy:**

Affirmed.

**Final Justification:**

The added experiments resolve several of my concerns and make the motivation clearer.

**Strength**
- The paper presents a simple but interesting idea.
- The additional results in the rebuttal are encouraging and strengthen the empirical case.
- Overall, the method appears potentially impactful.

**Remaining concerns**
-  The comparison between GRPO and OPSD can still be unfair in terms of generation length.
-  The training curves in the main paper appear not fully saturated, so it would be helpful to report the post-saturation performance mentioned in the rebuttal as well.

Overall, I find the paper reasonably sound but still with some evaluation gaps, and I have increased my score to borderline.

**Key Questions For Authors:**

Please address [W1]–[W5].

**Limitations:**

I recommend to add limitations which I discussed through [W1]-[W5].

**Strengths And Weaknesses:**

**Strengths**\
[S1] OPSD achieves comparable performance with other baselines while using fewer generated tokens in training time.

[S2] The method is straightforward and easy to understand.

---

**Weaknesses** \
[W1] The performance gain between GRPO and OPSD is quite small, which makes statistical reporting particularly important.

[W2] The context length used in the experiments appears too small for OPSD. While Qwen3-4B is native reasoning model which shows very long reasoning length (more than 32K), the reported reasoning length in Figure 4 is less than 4096 tokens for OPSD.

[W3] Also, for the same reason, the token-efficiency comparison between OPSD and GRPO in the right Figure 3 is unfair, because GRPO generates longer tokens during training (16K) while OPSD is capped in much shorter generation (2K). The paper should provide a comparison under a controlled setting. For example, by matching the generation length during training and inference.

[W4] As the paper has shown in Figure 3, the performance of both OPSD and GRPO does not appear to have been saturated, which suggests that the models need to be trained more.

[W5] The paper lacks sufficient component analysis. For instance, the suggested prompts are heuristically made by authors which should be verified through additional experiments.

---

> ### Author Rebuttal · Authors · 2026-03-31
>
> We thank the reviewer for the detailed feedback. Below we address each concern with additional experiments and clarifications.
>
> **W5: Insufficient component analysis; prompts are heuristically designed.**
>
> We agree that prompt design should be validated. To this end, we conducted systematic ablations along two dimensions.
>
> (1) we evaluate all four combinations of student/teacher generation modes (Thinking Mode on/off) by measuring per-token KL divergence, stratified into math, style, and other tokens:
>
> | Student | Teacher | Style KL | Math KL | Other KL |
> |---|---|---|---|---|
> | TM-off | TM-off | 0.68 | 0.12 | 0.11 |
> | TM-on  | TM-off | 0.51 | 0.10 | 0.17 |
> | TM-on  | TM-on  | 0.51 | 0.09 | 0.08 |
> | TM-off | TM-on  | 0.85 | 0.14 | 0.25 |
>
> The TM-off student / TM-on teacher configuration consistently yields the highest divergence on math-relevant tokens across model scales, providing a data-driven basis for our prompt design.
>
> (2) we ablate the amount of privileged info given to the teacher prompt. We compare full CoT supervision with answer-only supervision on 1.7b aime24:
>
> | | Base | Step 25 | Step 50 | Step 75 | Step 100 |
> |---|---|---|---|---|---|
> | CoT solution | 51.5% | 51.4% | 52.8% | 54.4% | 57.2% |
> | Answer only | 51.5% | 50.3% | 53.1% | 57.2% | 58.1% |
>
> Providing only the final answer still yields improvements, indicating that OPSD does not rely on full CoT and that the prompt design of privileged info is robust in this setup.
>
> **W1: Performance gain between GRPO and OPSD is small; statistical reporting is important.**
>
> With the improved TM-off/TM-on configuration, the gap between becomes larger and consistent across benchmarks even on 1b model.
>
>
>
> | Steps | AIME24 GRPO | AIME24 OPSD | \| | AIME25 GRPO | AIME25 OPSD | \| | HMMT25 GRPO | HMMT25 OPSD |
> |---|---|---|---|---|---|---|---|---|
> | 0 | 51.5 | 51.5 | \| | 36.7 | 36.7 | \| | 23.1 | 23.1 |
> | 25 | 51.1 | 51.4 | \| | 38.3 | 42.5 | \| | 24.2 | 24.7 |
> | 50 | 49.7 | 52.8 | \| | 36.4 | 43.9 | \| | 25.8 | 27.8 |
> | 75 | 50.3 | 54.4 | \| | 36.9 | 40.6 | \| | 23.6 | 26.9 |
> | 100 | 52.2 | 57.2 | \| | 37.8 | 41.1 | \| | 24.4 | 29.2 |
>
>
> The improvements also hold when using answer-only info. This consistency suggests the gains are systematic rather than due to statistical variance. Accuracies are averaged over 12 responses per problem, which also reduces per-problem variance.
>
> **W2: Context length for OPSD is too small.**
>
> The shorter generation length is due to we observe that increasing generation length from 1k to 4k does not yield consistent improvements. We hypothesize that earlier tokens means key reasoning branch points, while later tokens become increasingly predictable given long prefixes, resulting in weaker supervision. This is also noted in on-policy distillation blog from thinking machine labs in figure 6.(https://thinkingmachines.ai/blog/on-policy-distillation/). Evaluation at test time still uses the full 38k generation length.
>
> **W3: Token-efficiency comparison is unfair because GRPO generates longer sequences.**
>
> We would like to clarify that GRPO relies on full trajectories to obtain non-degenerate reward signals, but OPSD does not depend on verification rewards and therefore does not benefit from extending generation length in the same way. The shorter generation length in OPSD is not a constraint but a consequence of its design: it extracts supervision at the token level rather than from binary outcomes. Forcing OPSD to match GRPO’s 16k generation length would introduce additional compute without guaranteed improvements, as with our length ablation (1k–4k), where earlier tokens provide the most informative supervision.
>
> The observed difference in token usage is precisely the efficiency advantage of OPSD, which is a central goal of the method. To further illustrate the underlying cause of this gap, we measure the fraction of batches with zero reward variance in GRPO where all samples within a group is either all-correct or all-wrong.
>
> | Steps | Zero-Reward-Std Fraction (GRPO) |
> |---|---|
> | 10 | 0.65 |
> | 50 | 0.70 |
> | 60–100 | 0.65–0.95 |
>
> In contrast, OPSD provides dense token-level supervision at every step. The resulting token efficiency comparison is:
>
> | Steps | OPSD Tokens (M) | OPSD Avg% | GRPO Tokens (M) | GRPO Avg% |
> |---|---|---|---|---|
> | 0 | 0.00 | 37.1 | 0.00 | 37.1 |
> | 25 | 0.82 | 39.5 | 6.56 | 37.8 |
> | 50 | 1.64 | 41.5 | 13.12 | 37.3 |
> | 75 | 2.46 | 40.6 | 19.67 | 36.9 |
> | 100 | 3.28 | 42.5 | 26.23 | 38.1 |
>
> OPSD achieves higher accuracy with substantially fewer tokens, which directly reflects its dense supervision signal.
>
> **W4: Models do not appear saturated; more training is needed.**
>
> We extended training and observe that GRPO does not improve further and often degrades due to entropy collapse. In contrast, OPSD maintains or increases student entropy.
>
> We hope these new results and clarifications address your concerns. We are happy to address any further questions you might have.

---

> > ### Author Rebuttal · Reviewer_U4bM · 2026-04-03
> >
> > Thank you for the thoughtful rebuttal. The added experiments resolve several of my concerns and make the motivation clearer.
> >
> > I still think the paper would be stronger with (i) a controlled comparison at shorter GRPO rollout lengths (4K), since the claim that early tokens matter may also apply to GRPO, (ii) it would also help to report the post-saturation accuracies mentioned in the rebuttal.
> >
> > While some concerns about the controlled comparison and completeness of reporting remain, the core idea is clear and potentially impactful. For these reasons, I have increased my score to borderline for now.

---

> > > ### Author Response · Authors · 2026-04-07
> > >
> > > Thank you for the constructive feedback and for the positive update in score — we are glad the additional experiments clarified the motivation.
> > >
> > > Regarding the suggestion on (i) Controlled comparison with shorter GRPO rollouts (4K).
> > > We agree that controlling rollout length is a useful comparison. However, 4k generation length would degrade GRPO's performance as we find that rollout length of 4K results in many trajectories not reaching a verifiable final answer (mean completion lengths on this dataset are ~10K–14K) and advantages can not be calculated properly. This highlights the difference that GRPO relies on trajectory-level rewards that need final answer generation from longer rollouts, whereas OPSD provides dense token-level supervision and does not depend on reaching final answers. We will clarify this point more explicitly in the revision.
> > >
> > > Regarding the suggestion on (ii) Post-saturation performance.
> > > We agree that reporting post-saturation behavior is important. We observe that GRPO tends to degrade in later stages due to entropy collapse, particularly for smaller models (e.g., 1.7B). Therefore, we have reported the best achieved performance for GRPO in the current paper. We will include post-saturation curves in figure in the revision for completeness.
> > >
> > > Thanks again for the constructive feedbacks which have meaningfully improved our paper, we will incorporate the above results (e.g ablations on prompts, privileged information density, wider performance gap with improved prompting, and GRPO's post-saturation performance) into the paper to enhance the paper.

---

### Official Review · Reviewer_ebs6 · 2026-03-12

**Soundness:** 3
**Presentation:** 4
**Significance:** 3
**Originality:** 3
**Overall Recommendation:** 4
**Confidence:** 4

**Summary:**

This paper introduces On-Policy Self-Distillation, a learning paradigm where the teacher leverages privileged information (such as the reference solution) and provides learning signal to the student version of the same model via reverse KL. The authors demonstrate that OPSD achieves comparable performance to GRPO while being more efficient in terms of training sample and token efficiency.

**Compliance With Llm Reviewing Policy:**

Affirmed.

**Final Justification:**

The rebuttal addressed my main concerns, but the experiments in this paper still lacks transparency and persuasiveness. In particular:
- The very short generation length is abnormal for Qwen3 thinking models (with thinking mode on), both in training setup and in evaluation generation length.
- The observed improvements are marginal according to Table 2.
- The distillation is based on full vocabulary logits, which is expensive. Without full vocabulary logits, improvement diminishes.

With these weaknesses, I decide not to increase my score.

**Key Questions For Authors:**

1. Can you show any actual teacher generated reasoning traces after seeing the reference answer? My concern is that teachers may tend to leak answers before even completing the reasoning process, which could affect the student's distilled behavior.
2. Could you evaluate whether teacher model comprehends the reference solution, e.g. via masking the final answer part in reference CoT and measuring the teacher's accuracy given this partial reference?
3. Why use a very short generation length (<=4k) for OPSD? Can you provide results for 16k runs? Could you also report generation lengths for GRPO and OPSD on benchmarks in Table 2?
4. Can you provide results for OPSD with teacher being updated along the student? Why would this configuration cause instability?
5. Since OPSD relies on reference solutions with CoT, which are generated by more powerful LLMs or curated by other means, is OPSD implicitly OPD from a more powerful model? Would this limit the potential compared to GRPO at larger scales?

**Limitations:**

yes

**Strengths And Weaknesses:**

### Strengths
1. This paper is well written and easy to follow.
2. The method and experiments are straightforward and shows positive results.


### Weaknesses
1. The experiments and discussions are not comprehensive enough. Additional details that I would like to know are listed in the Questions below.
2. Lack comparison to On-Policy Distill from a more powerful model (e.g. OPD from the model generating reference solutions for OPSD).
3. The improvements in accuracy seemed marginal according to Table 2.

---

> ### Author Rebuttal · Authors · 2026-03-31
>
> We thank the reviewer for the detailed and constructive feedback. We appreciate your recognition of the paper's presentation quality and positive empirical results. Below we address each concern.
>
> **W1/Q1: Can you show actual teacher-generated reasoning traces? Concern about answer leakage before reasoning.**
>
> Below are two actual teacher-generated traces (truncated for space). For the problem of finding the maximum of $(2a^3+27c-9ab)\lambda^3$:
>
> > *"Let me recall Vieta's formulas for cubic equations... So, substituting x = y − a/3 into the polynomial, we get... Expanding this, the coefficient of y² is... let me compute the constant term..."*
>
> And for a simpler problem:
>
> > *"First, I remember that when comparing exponents with the same exponent, the larger base gives the larger value. But here the exponents aren't the same, so I need to adjust them... For p, 2^3009, since 3009/3 = 1003, that's (2³)^1003 = 8^1003. Similarly q = (3³)^1003 = 9^1003... so comparing 8, 9, 5: since 5 < 8 < 9 we get r < p < q..."*
>
> Both traces open with exploratory words ("let me", "I remember") and arrive at the answer only after working through the derivation. We address the leakage concern by designing the teacher's transition prompt explicitly instructs the teacher to *"not copy or paraphrase the reference solution, and instead derive the same final answer using independent reasoning."* This discourages verbatim copying and encourages re-derivation.
>
> **Q2: Could you evaluate teacher comprehension by masking the final answer in the reference CoT and measuring teacher accuracy?**
>
> Thank you for the suggestion. We evaluate this by masking the final answer in the reference CoT and instructing the model to derive its own answer from the remaining reasoning steps, tested on 200 examples from the training dataset:
>
> | Model | Accuracy (answer masked) |
> |---|---|
> | Qwen3-1.7B | 75.0% |
> | Qwen3-4B | 84.5% |
>
> Both models can meaningfully comprehend the reference reasoning, with larger models doing so more reliably,
>
> **Q3: Why use short generation length (≤4k)? Please provide 16k results and also report generation lengths for GRPO and OPSD on Table 2 benchmarks.**
>
> We used short generation lengths because in our experiments we find that 1k-2k already yield comparable gains, with diminishing returns beyond that. We hypothesize two reasons: (1) earlier tokens often correspond to critical reasoning branch points that are most informative for distillation, while (2) later tokens become increasingly predictable to the teacher even when the student's reasoning is going wrong, providing a weaker and noisy learning signal, which is also noted from the Thinking Machines Lab OPD blog post (Figure 6). In Table 2, OPSD used 2k generation length while GRPO used 16k generation length.
>
> **Q4: Can you provide results for OPSD with the teacher updated jointly with the student? Why does this cause instability?**
>
> We experimented with this and found that it leads to training instability and performance degradation consistently. We think the cause is non-robustness to student mistakes, if student learns from a noisy trace, it will quickly degenerate (for example generating random strings), which is easy to hack by mimicking. Like in RLVR GRPO we have reference policy KL to stablize training, fixing the teacher in OPSD also acts as an implicit regularizer as it stabilizes the distillation target and prevents the teacher from drifting toward distributions that rationalize the student's errors rather than correcting them.
>
> **Q5: Since OPSD relies on CoT from more powerful LLMs, is it implicitly OPD from a stronger model? Does this limit its potential compared to GRPO at larger scales?**
>
> This is a good point, and we address it from two angles. First, OPSD does not strictly require full CoT — we ran a new experiment on Qwen3-1.7B comparing full CoT versus answer-only as privileged context:
>
> | | Base | Step 25 | Step 50 | Step 75 | Step 100 |
> |---|---|---|---|---|---|
> | CoT solution | 51.5% | 51.4% | 52.8% | 54.4% | 57.2% |
> | Answer only | 51.5% | 50.3% | 53.1% | 57.2% | 58.1% |
>
> Providing only the final answer still yields consistent improvements, confirming that OPSD does not depend on externally generated reasoning traces. The model can rationalize toward the correct answer even from a minimal hint, similar in spirit to STaR [1] — though unlike STaR, OPSD performs on-policy distribution matching rather than offline SFT on generated rationales.
>
> Second, even when full CoT from a stronger model is used, the supervision signal in OPSD is not directly the stronger model's distribution, it is the student's own rationalization of that CoT, which is more in-distribution than standard off-policy distillation, and may reduce the risk of catastrophic forgetting that can arise from purely offline data [2].
>
> [1] Zelikman et al., STaR: Bootstrapping Reasoning with Reasoning.
>
> [2] https://arxiv.org/pdf/2509.04259

---

> > ### Author Rebuttal · Reviewer_ebs6 · 2026-04-01
> >
> > I thank the authors for addressing all my questions. I will keep my score.

---

### Official Review · Reviewer_GXYv · 2026-03-12

**Soundness:** 3
**Presentation:** 2
**Significance:** 3
**Originality:** 2
**Overall Recommendation:** 4
**Confidence:** 4

**Summary:**

The paper focuses on post-training. It proposes a new method OPSD, which combines OPD and self-improvement. Unlike the previous OPD, which typically relies on a stronger teacher model, it employs a self-teaching strategy, which means it takes itself as a teacher model by adding groundtruth into the prompt to predict the next token. The key advantage is that it can enhance the sampling efficiency and the signal is dense, which in some sense is a prm reward.

**Compliance With Llm Reviewing Policy:**

Affirmed.

**Final Justification:**

The rebuttal addresses my concerns. I maintain my score.

**Key Questions For Authors:**

1. Baseline Comparison: When we use GRPO to train the model, we do not need the Cot. However, when we use OPSD, we need to know the Cot. Do you have the experiments if we only know $y^*$ by using OPSD?
2. While the authors say OPSD can improve the sampling cost, OPSD needs memory costs since you need to store the full vocabulary distribution. Later I found you use Sampled-token distillation, now my question is, do you compare this in your Table 2?

**Limitations:**

yes

**Strengths And Weaknesses:**

Strengths:
1. The paper is well-organized.
2.  The topic is very good.
3. Combining OPD and self-improvement is pretty interesting.

Weakness:
1. See the questions.

---

> ### Author Rebuttal · Authors · 2026-03-31
>
> We thank reviewer GXYv for the constructive feedback and for recognizing the novelty of combining on-policy distillation with self-improvement. We address the questions below.
>
> **Q1: Do you have experiments using only the final answer y\* (without CoT) in OPSD?**
>
> Yes, we ran this comparison. The table below shows average@12 accuracy on Qwen3-1.7B, comparing two privileged context conditions: the full CoT solution (step-by-step reasoning) versus only the final boxed answer. For the answer-only condition, we changed the transition prompt of the teacher to be "You are told the correct final answer above. Now derive it yourself from scratch using your own reasoning..." to encourage the model to rationalize only when given the concise answer.
>
> | | Base | Step 25 | Step 50 | Step 75 | Step 100 |
> |---|---|---|---|---|---|
> | CoT solution | 51.5% | 51.4% | 52.8% | 54.4% | 57.2% |
> | Answer only | 51.5% | 50.3% | 53.1% | 57.2% | 58.1% |
>
> Providing only the final answer still yields improvements over the base model, this shows that OPSD does not strictly require CoT. When the problem is within the model's reasoning capacity, the model can rationalize toward the correct answer even when conditioned on a single scalar hint. We think this is similar to STaR [1], which also conditions only on a hint or answer, though STaR performs offline SFT on generated rationales rather than on-policy distribution matching.
>
> **Q2: Does Table 2 compare sampled-token distillation vs. full-vocabulary distillation?**
>
> Table 2 reports results using full-vocabulary logit distillation only. The direct comparison between the two objectives appears in Table 3, where we ablate on Qwen3-4B across two benchmarks (AIME25 and HMMT25). Full-vocabulary distillation outperforms sampled-token distillation on AIME25 and HMMT25 (60.0% vs. 57.3%). We chose full-vocabulary as our default based on these findings and the prior results of GKD; due to compute constraints we did not extend this ablation to all model scales and tasks in Table 2. We will clarify this more explicitly in the table caption.
>
> Regarding the memory concern: while full-vocabulary distillation does require storing vocabulary-sized logits at each token position, OPSD's short generation budget (1k–2k tokens, versus GRPO's 16k) keeps the absolute memory overhead manageable. The reason why the short sequence length of 1k-2k works for OPSD is because that the earlier tokens are more central for reasoning, as some branching tokens can happen earlier and for later tokens, even when the tokens are wrong the teacher might not have high penalty because the teacher is already conditioned on very long student generation, and the wrong tokens becomes very predictable to the teacher now. This is also noted in on-policy distillation blog post from thinking machine labs in figure 6.(https://thinkingmachines.ai/blog/on-policy-distillation/). Therefore we think the memory overhead compared to OPSD's efficiency is manageable, in practice, on 8×A100s, OPSD finishes training in under 30 minutes. The prior work GKD (Agarwal et al., 2024), which our method builds on, also uses full-vocabulary distillation and demonstrated consistent advantages over sampled-token approaches.
>
> [1] STaR: Bootstrapping Reasoning With Reasoning. https://arxiv.org/abs/2203.14465

---

> > ### Author Rebuttal · Reviewer_GXYv · 2026-03-31
> >
> > The rebuttal properly addresses my concerns. I maintain my score.

---

### Official Review · Reviewer_v4ui · 2026-03-13

**Soundness:** 3
**Presentation:** 3
**Significance:** 3
**Originality:** 3
**Overall Recommendation:** 5
**Confidence:** 4

**Summary:**

The paper introduces On-Policy Self-Distillation, a post-training framework designed to improve the reasoning capabilities of LLMs without relying on a separate, larger teacher model. The teacher uses privileged information like ground-truth solutions to provide dense token-level supervision. The authors demonstrate that OPSD outperforms SFT and achieves comparable or superior mathematical reasoning performance to GRPO on benchmarks like AIME and HMMT, while requiring significantly fewer generated tokens during training.

**Compliance With Llm Reviewing Policy:**

Affirmed.

**Final Justification:**

My concerns have been addressed and I maintain my positive score.

**Key Questions For Authors:**

- How does the performance of OPSD change if the privileged context provided to the teacher is shortened or less detailed?
- Is there a specific model scale threshold below which the teacher fails to provide meaningful rationalization for the student?
- How does OPSD compare directly with SFT on the same privileged CoT traces in terms of final reasoning accuracy?

**Limitations:**

Yes

**Strengths And Weaknesses:**

**Strengths**

- Achieves significantly higher token efficiency than traditional RL methods like GRPO by providing dense feedback instead of sparse rewards.
- Eliminates the need for a separate larger teacher model or complex PRM training, making the setup more resource-efficient
- Reduces exposure bias by training on the student's own generated trajectories, ensuring better distribution alignment.

**Weakness**

- Since the teacher and student share parameters, the teacher's capability is bounded by the model's inherent reasoning limits even with privileged context
- Evaluation is primarily focused on mathematical reasoning, leaving the generalizability to other complex domains like coding or creative writing unproven.
- The method relies heavily on high-quality CoT reasoning traces in the training set to serve as effective privileged information.

---

> ### Author Rebuttal · Authors · 2026-03-31
>
> We sincerely thank the reviewer for the thorough and positive evaluation. We are glad you found OPSD's token efficiency gains and elimination of a separate teacher model compelling. Below we address each question with clarifications and new experiment.
>
> **Q1: How does OPSD performance change if the privileged context provided to the teacher is shortened or less detailed?**
>
> Thanks for this question, we think if the model is within capacity to understand the less detailed solution (for example, just the final answer), it can still rationalize and provide meaningful supervision signal. To test this, we ran experiment on Qwen3-1.7B comparing two forms of privileged information: (1) full CoT solution containing detailed step-by-step reasoning, and (2) answer-only providing just the final boxed answer. Results are shown below. For the answer-only condition, we changed the transition prompt of the teacher to be "You are told the correct final answer above. Now derive it yourself from scratch using your own reasoning..." to encourage the model to rationalize only when given the concise answer.
>
> | | Base | Step 25 | Step 50 | Step 75 | Step 100 |
> |---|---|---|---|---|---|
> | CoT solution | 51.5% | 51.4% | 52.8% | 54.4% | 57.2% |
> | Answer only | 51.5% | 50.3% | 53.1% | 57.2% | 58.1% |
>
> Providing only the final answer still yields improvements over the base model, this shows that OPSD does not strictly require CoT. When the problem is within the model's reasoning capacity, the model can rationalize toward the correct answer even when conditioned on a single scalar hint. We think this is similar to STaR [1], which also conditions only on a hint or answer, though STaR performs offline SFT on generated rationales rather than on-policy distribution matching.
>
> **Q2: Is there a specific model scale threshold below which the teacher fails to provide meaningful rationalization?**
>
> We find the answer depends not only on model scale but on a combination of factors: problem difficulty, the off-policy-ness of the CoT data relative to the student's generation style, and the model's general comprehension of the provided solution. while gains grow progressively at 4B and 8B. This is consistent with larger models having stronger rationalization capacity, enabling the teacher to produce more informative distributions. Rather than a hard threshold, we observe a soft capacity dependence: as model size decreases, rationalization quality degrades gracefully, and the distillation signal becomes noisier but remains useful. We flag in our limitation that evaluation beyond 8B as an important open direction.
>
> **Q3: How does OPSD compare directly with SFT on the same privileged CoT traces in terms of final reasoning accuracy?**
>
> Table 2 has this comparison. SFT on the same CoT traces can sometime degrades performance relative to the base model. We attribute this to a mismatch in reasoning style: the solutions style can be concise and direct and maybe very out-of-distribution style to the current Qwen3 models, lacking the backtracking and exploratory behaviors (e.g., "wait," "let me reconsider") that Qwen3 thinking-mode models rely on extensively at inference. Fine-tuning on these traces causes the model to reduce its generation length at test time which hurts the base model's strong performance. OPSD avoids this by rather than fine-tuning on these traces, it uses them as privileged teacher context to supervise the student's own on-policy rollouts generated in the student's own reasoning style.
>
> [1] STaR: Bootstrapping Reasoning With Reasoning. https://arxiv.org/abs/2203.14465

---

> > ### Author Rebuttal · Reviewer_v4ui · 2026-04-02
> >
> > My concerns have been adequately addressed. I will keep my postive score.

---

### Decision · Program_Chairs · 2026-04-30

**Decision:**

Accept (regular)

**Comment:**

This paper proposes On-Policy Self-Distillation (OPSD), a highly efficient post-training framework for LLMs. Instead of relying on a separate, larger teacher model or sparse reward RL (like GRPO), OPSD uses a single model as both teacher and student. By conditioning the teacher policy on privileged information and the student policy on the standard prompt, the model provides dense, token-level KL-divergence supervision to its own on-policy rollouts. This yields competitive mathematical reasoning capabilities with a fraction of the generated tokens required by standard RL.

Strengths:

Exceptional token and resource efficiency: By providing dense per-token feedback, OPSD achieves comparable or superior performance to GRPO while using roughly 8x fewer generated tokens during training. It also eliminates the need to host a separate, massive teacher model.

Elegant paradigm: The framework seamlessly blends on-policy distribution matching with self-improvement, avoiding the exposure bias of offline SFT and the sparse-reward bottleneck of RLVR.

Thorough rebuttal: The authors provided excellent responses to reviewer questions, notably proving that OPSD remains effective even when the teacher only sees the final answer (rather than a full CoT trace) and clearly justifying why short rollouts work for OPSD but fail for GRPO.

Weaknesses:

Memory Cost: Full-vocabulary logit distillation is memory-intensive, though the authors rightly argue this is mitigated by the short generation lengths (1k-2k tokens) needed for OPSD.

Scope and Marginal Gains: The evaluations are confined strictly to mathematical reasoning, and the absolute performance gains over standard GRPO, while achieved much faster, are somewhat modest.

The idea of using privileged context to turn a model into its own dense-reward teacher is both intuitive and empirically validated. The reviewers recognized the novelty and efficiency of the approach, and the authors successfully defended their design choices (such as rollout lengths and prompt structures) during the rebuttal. Given its high relevance to the current LLM post-training landscape, this paper is a solid accept for ICML.